# Discovery of Novel TASK-3 Channel Blockers Using a Pharmacophore-Based Virtual Screening

**DOI:** 10.3390/ijms20164014

**Published:** 2019-08-17

**Authors:** David Ramírez, Guierdy Concha, Bárbara Arévalo, Luis Prent-Peñaloza, Leandro Zúñiga, Aytug K. Kiper, Susanne Rinné, Miguel Reyes-Parada, Niels Decher, Wendy González, Julio Caballero

**Affiliations:** 1Instituto de Ciencias Biomédicas, Facultad de Ciencias de la Salud, Universidad Autónoma de Chile, El Llano Subercaseaux 2801-Piso 5, 8900000 Santiago, Chile; 2Centro de Investigaciones Médicas (CIM), Programa de Investigación Asociativa en Cáncer Gástrico (PIA-CG), Escuela de Medicina, Universidad de Talca, 2 Norte 685, 3460000 Talca, Chile; 3Informática y Telecomunicaciones, Universidad Tecnológica de Chile Inacap, Sede Talca, Av. San Miguel 3496, 3460000 Talca, Chile; 4Organic Synthesis Laboratory and Biological Activity (LSO-Act-Bio), PhD Applied Sciences, Faculty of Engineering, Institute of Chemistry of Natural Resources, Universidad de Talca, 1 Poniente No. 1141, 3460000 Talca, Chile; 5Institute for Physiology and Pathophysiology, Vegetative Physiology, Philipps-University of Marburg, Deutschhausstraße 2, 35037 Marburg, Germany; 6Centro de Investigación Biomédica y Aplicada (CIBAP), Escuela de Medicina, Facultad de Ciencias Médicas, Universidad de Santiago de Chile, Av. Libertador Bernardo O’Higgins 3677, 8900000 Santiago, Chile; 7Facultad de Ciencias de la Salud, Universidad Autónoma de Chile, Avenida Pedro de Valdivia 425, 8900000 Santiago, Chile; 8Center for Mind, Brain and Behavior (CMBB), University of Marburg and Justus Liebig University Giessen, 35037 Marburg, Germany; 9Centro de Bioinformática y Simulación Molecular (CBSM), Universidad de Talca. 1 Poniente No. 1141, 3460000 Talca, Chile; 10Millennium Nucleus of Ion Channels-Associated Diseases (MiNICAD), Universidad de Talca, 3460000 Talca, Chile

**Keywords:** TASK-3 channel, drug design, TASK channels blockers, pharmacophore-based virtual screening, lead optimization

## Abstract

TASK-3 is a two-pore domain potassium (K_2P_) channel highly expressed in the hippocampus, cerebellum, and cortex. TASK-3 has been identified as an oncogenic potassium channel and it is overexpressed in different cancer types. For this reason, the development of new TASK-3 blockers could influence the pharmacological treatment of cancer and several neurological conditions. In the present work, we searched for novel TASK-3 blockers by using a virtual screening protocol that includes pharmacophore modeling, molecular docking, and free energy calculations. With this protocol, 19 potential TASK-3 blockers were identified. These molecules were tested in TASK-3 using patch clamp, and one blocker (**DR16**) was identified with an IC_50_ = 56.8 ± 3.9 μM. Using **DR16** as a scaffold, we designed **DR16.1**, a novel TASK-3 inhibitor, with an IC_50_ = 14.2 ± 3.4 μM. Our finding takes on greater relevance considering that not many inhibitory TASK-3 modulators have been reported in the scientific literature until today. These two novel TASK-3 channel inhibitors (**DR16** and **DR16.1**) are the first compounds found using a pharmacophore-based virtual screening and rational drug design protocol.

## 1. Introduction

Two-pore domain potassium (K_2P_) channels have a major role in the regulation of cell excitability and membrane potential in excitable and non-excitable cells [1]. K_2P_ channels contain two pore-forming loops and four transmembrane domains per subunit, creating dimeric channels [2]. The TASK (tandem of pore domains in a weak inwardly rectifying K^+^ channel [TWIK]-related acid-sensitive K^+^ channel) subfamily is integrated by TASK-1 [3], TASK-3 [4] (sharing 58.9% of aminoacidic (aa) sequence identity [5]), and TASK-5 (sharing 51.4 % and 55.1% of aa sequence identity with TASK-1 and TASK-3, respectively) [6]. Some studies have demonstrated that TASK channels participate in the chemical control of breathing due to their intrinsic pH and O_2_ sensitivity [7,8,9]. These channels are expressed in the nervous, cardiovascular, genitourinary, and gastrointestinal systems [10]. They are involved in chemosensation [11] and also have a role in the regulation of the immune system [12]. TASK channels are acid-sensitive and anesthetic-activated members of the K_2P_ family. They contribute to the effects of general anesthetics due to the activation of background K^+^ currents causing a decrease of excitability by neuronal hyperpolarization [13], which makes these channels prominent molecular targets for these drugs.

K_2P_ channels exhibit a different topology and 3D-structure in relation with other K^+^ channels. The recently released crystallographic structures of some K_2P_ channels, such as TRAAK (PDBs:3UM7 [5] and 4I9W [14]), TREK-1 (PDB:4TWK, 6CQ6 and 6CQ8 [15]), TREK-2 (PDBs:4BW5, 4XDJ, 4XDK and 4DKL [16]), and TWIK-1 (PDB:3UKM [17]), show that these proteins exhibit some features that characterize their unique gating and ion permeation properties. For example, close to the membrane center, the TM2 helix bends approximately 20°. This structural change generates two side-cavities named fenestrations, connecting the pore to the hydrophobic core of the membrane [5,17]. These fenestrations have a key role in the modulation of K_2P_ channels [18], providing binding pockets for drugs like norfluoxetine in TREK-2 [16] and bupivacaine in TASK-1 [19]. It has been proven by molecular dynamic simulations that drugs like A1899 and PK-THPP bind preferentially to TASK-1 and TASK-3 channels with open fenestrations, respectively [18,20]. Thus, these hydrophobic cavities are potential drug-binding sites, and also might provide new pathways that could guide blockers into their binding site.

TASK-3 is highly expressed in the hippocampus, cerebellum, and cortex [21], and some previous studies have described that TASK-3 regulates neurotransmitter function [22]. The development of new selective TASK-3 modulators could influence the pharmacological treatment of several neurological conditions, such as sleep disorders, neurodegeneration, cognitive impairment, Huntington’s disease, Parkinson’s disease, or major depressive disorder [23]. 

Not many promising inhibitory TASK-3 modulators have been reported in the scientific literature [24]. One of the few reports in this context was made by Coburn et al. [25]; they informed the use of aminopyrimidine derivatives as potent TASK-3 blockers. Noriega-Navarro et al. [26] reported the application of dihydropyrrolo[2,1-a]isoquinoline derivatives (DPIs) as novel TASK inhibitors. In this sense, the use of fused heterocyclic-compounds has attracted attention as new TASK modulators. Therefore, the development of simple theoretical/experimental methodologies is necessary to find new compounds with a different chemical nature with potential usefulness as TASK-3 modulators. 

In this study, we developed a systematic pipeline (Scheme 1) to search novel TASK-3 blockers that includes a pharmacophore-based virtual screening (*e*-PBVS). The 19 putative blockers found were screened against human TASK-3 by a patch clamp, obtaining one active ligand that exhibited inhibitory activity against TASK-3 in the μM range. The active ligand was used as a scaffold and a new compound was designed, synthetized, and tested against TASK-3, exhibiting a four-fold higher activity.

## 2. Results

### 2.1. TASK-3 Modeling

The TASK-3 aminoacidic sequence shares 27.2% identity with TWIK-1, 23.7% with TRAAK, and 26.2% with TREK-2 [5]; hence, all three K_2P_ channels 3D-structures are acceptable as templates to build homology models [27]. TASK-3 models (Table 1) were inserted in a biological membrane and subjected to 10 ns MD simulations. Models were stabilized before 5 ns, which is analyzed from the RMSDs (Root Mean Square Deviation) of the backbone atoms as a function of the simulation time for the models using their initial configurations as references (Appendix AA). The stabilized models were validated using Procheck and ProSA. Both validations showed that TASK-3 homology models have a good quality. They exhibit more than 90% of the residues in the most favored regions of the Ramachandran plots [28] (Appendix A), and the z-score in the same range that experimentally determined structure proteins of the Protein Data Bank (PDB) [29] (Appendix A).

The different fenestration states were further analyzed in the T3-trCO model because it has the close-open fenestration state. Some residues identified by Streit et al. [30] and our group as key residues for the interaction of drugs, such as A1899 and PK-THPP, with TASK-1 and TASK-3 channels [18,20] are shown in Appendix A (these residues are surrounded by a transparent surface to detail the closed and open fenestrations, at the left and the right, respectively). The TM4 segment of chain A is further to the TM2 segment of chain B, resulting in the opening of the fenestration as described by Aryal et al. [31]. It can be seen in Appendix A how the hydrophobic side chain of Leu239 (yellow) at the TM4 segment is oriented to the hydrophobic fenestration cavity interacting with Leu197 (green) at the TM2 segment. Ile235 (purple) at the TM4 segment and Val115 (white) of the inner helix 1 are also establishing non-bonding interactions.

### 2.2. e-Pharmacophore Modeling

Twelve TASK-3 blockers were selected to build an energy-optimized pharmacophore (*e*-Pharmacophore). They have an IC_50_ range of 0.035 to 160 μM (Table 2). Ensembles that contain three pharmacophore points were identified, with hydrogen bond acceptors (*A*), hydrophobic groups (*H*), and aromatic rings (*R*). The *e*-Pharmacophore hypotheses were scored using the phase scoring function (Table 3). The top scored pharmacophore hypothesis was a three-point pharmacophore ensemble with two hydrogen bond acceptors (features *A1* and *A2*) and one aromatic ring (feature *R*). The score measures how well the vectors from the pharmacophore features are aligned in the structures that contribute to the hypothesis, when the structures themselves are aligned to the pharmacophore. On the other hand, the selectivity value estimates the rarity of the hypothesis, based on the World Drug Index; this term is the negative logarithm of the fraction of molecules in the index that matches the hypothesis. Thus, the lower the selectivity value, the less molecules exhibit the pharmacophore. A three-dimensional (3D) representation of the pharmacophore hypothesis number one is shown in Figure 1, with six TASK-3 blockers—12f (cyan), 23 (green), 17e (orange), A1899 (yellow), GW2974 (pink), and Loratadine (gray)—adopting the conformations that fit in the model. It can be seen how the geometry of hypothesis number one is conserved in the blockers. For compounds 12f, 17e, and 23, the *A1* feature is the carbonyl oxygen, *A2* is the N3 of the 5,6,7,8-tetrahydropyrido[4,3-*d*]pyrimidine, and *R* is the phenyl moiety bound to the carbonyl group. For A1899, *A1* and *A2* correspond to carbonyl oxygens and *R* is the phenyl group of the methoxyphenyl substituent. For GW2974, *R* is the phenyl of the 1*H*-indazole and *A1* and *A2* correspond to N3 and N7 of the pyrido[3,4-*d*]pyrimidine group. For loratadine, *A1* is the nitrogen of the pyridine, *A2* is the ether oxygen of the carboxylate group, and *R* is the chlorophenyl group. We analyzed the local charges of atoms of the *A1* and *A2* groups and we observed that they have highly negative Mulliken atomic charges. The general site measurements of the *e*-Pharmacophore model (distances and angles between the features) are given in Appendix A.

### 2.3. Virtual Screening

The *e*-Pharmacophore hypothesis in conjunction with the four TASK-3 models were used as input for *e*-PBVS using ZINCPharmer pharmacophore search software [33]. The search explored more than 200 million conformations from more than 22 million compounds of the ZINC (purchasable) database [34]. Database hits were ranked according to RMSD (see Section 4.3), resulting in 5000 hits for each TASK-3 model, with a total of 20,000 hits.

After the ‘Molecular docking 4’ re-docking process (Scheme 1) with Glide XP and the implementation of stage two of the protocol shown in Scheme 1, the predicted hit ligands from the ZINC database were identified. The 19 ligands (DR1–DR19) with the lowest MM-GBSA (Molecular Mechanics combined with the Generalized Born and Surface Area) binding free energies are listed in Table 4. The hit ligands interact with at least two of the four TASK-3 models developed in this study, with different $\Delta G_{\mathrm{Bind}}$ energies. These hits share several chemical features among them, such as amide moieties, aromatic rings, and heterocycles, and hydrogen bond acceptor groups, such as oxygen and nitrogen atoms (Appendix A). These chemical features are also found in the TASK-3 blockers reported in the literature. 

### 2.4. Biological Activities of Identified Hits

The 19 hits were screened against TASK-3 using patch-clamp. The results of these evaluations were no activity for the majority of compounds, but compound **DR16** was identified as an active ligand, with an IC_50_ = 56.8 ± 3.9 μM. In Figure 2A, the conserved pharmacophore among the reported TASK-3 blockers (Figure 1) is illustrated in **DR16**, along with its dose–response curve against TASK-3 (Figure 2B) and TASK-1 (Figure 2C) channels. The hit compounds identified from the ZINC database and selected for the experimental activity evaluations were acquired from the following suppliers: AKos Consulting & Solutions Deutschland GmbH (Steinen, Germany): **DR1**-**DR10**; Ambinter c/o Greenpharma (Orléans, France): **DR11** and **DR12**; EnamineStore Ltd. (Kyiv, Ukraine): **DR13**-**DR16**; and Vitas-M Limited (Hong Kong, China): **DR17**-**DR19**. All compounds were solubilized in DMSO (dimethyl sulfoxide) 10 mM stock.

### 2.5. Binding Model of DR16

The proposed binding modes of **DR16** in the TASK-3 channels with different fenestration states predicted by our protocol are shown in Figure 3. It is noticeable that different orientations were predicted for **DR16** in the T3-treCC and T3-twiOO models. We found two possible binding sites for **DR16**, one at the inner cavity in T3-treCC (Figure 3A,B) and the other at the fenestration in T3-twiOO (Figure 3C,D). However, according to the $\Delta G_{Bind}$ energies of **DR16** in both models (Table 4), the binding between **DR16** and the fenestration at the T3-twiOO model is more favorable ($\Delta G_{Bind}$ = −55.89 kcal/mol).

The binding mode of **DR16** inside the T3-treCC model is characterized by the presence of two hydrogen bonds between the carbonyl oxygen of the amide group of the ligand and the side chain OH groups of the residues, Thr93 (chain B) and Thr199 (chain A) (Figure 3A,B). **DR16** also presents, in the obtained conformation inside the T3-treCC model a hydrogen bond between the OH of the ligand and backbone of the residue Leu197, and a π–π stacking interaction with the Phe125 (Figure 3A,B). It is important to notice that the interactions established by **DR16** with T3-treCC involve two of the three pharmacophoric descriptors found for TASK-3 channel blockers (Figure 3A,B), the aromatic ring and a H-bond acceptor (Figure 2A). The OH moiety is interacting as an H-bond donor. Phe125 was reported as a putative false positive binding residue for A1899 because the docking pose of A1899 predicted this residue as part of the binding site, but the experimental data did not fit with those results in TASK-1 [30]. 

In the T3-twiOO–**DR16** complex, the ligand is located inside the fenestration and the OH of the ligand is oriented towards the central cavity, interacting through a hydrogen bond with the backbone CO of the Leu232 (Figure 3C,D). The NH of the ligand also establishes a hydrogen bond with the backbone CO of the Leu197. Besides, the benzofuran of the ligand forms a π–π stacking interaction with the residue Phe194 (Figure 3C,D). Finally, it can be seen in the T3-twiOO–**DR16** complex that hydrophobic interactions between the ligand and the residues Val115, Ile118, Pro119, Leu122, Leu171, Ile235, and Leu239 (Figure 3C,D) located at the fenestrations occur. In this complex, only the aromatic ring pharmacophoric feature of **DR16** is interacting with the channel, and the two hydrogen bonds interactions do not fit with the pharmacophore described previously. However, these interactions are present between the TASK-3 channel and the two moieties, which can behave like an H-bond acceptor or donor, making the interaction possibilities of this blocker versatile.

The different binding modes of **DR16** in the two different models of TASK-3 with the fenestration in the closed (T3-treCC) and open state (T3-twiOO) allow us to suggest that the hydrophobic moieties could interact with the fenestration hydrophobic residues when TASK-3 has the fenestrations in the open state, just like A1899 [18] or bupivacaine [19] interacts with TASK-1 channels.

### 2.6. Drug Design of Novel TASK-3 Blockers Using DR16 As a Scaffold

A novel compound was designed by using the knowledge of the TASK blockers reported in the literature, specially A1899 [18,30,35]. We used the structure of **DR16** as the starting point. **DR16** was expanded by adding a 4-hydroxybenzamide group at position four of the benzofurane. With this modification, we added an aromatic ring and hydrogen bond donor/acceptors, and we obtained the novel compound **DR16.1**, which is similar to the distal anisole moiety linked by an amide group in the A1899 blocker.

The computational model of the compound **DR16.1** forming a complex with TASK-3 was constructed by docking (Figure 4). The binding affinity of the novel compound against the T3-twiOO model by MM-GBSA was evaluated. The docking energy obtained for **DR16.1** was −8.137 kcal/mol, and the MM/GBSA $\Delta G_{\mathrm{Bind}}$ energy was −89.02 kcal/mol, with a remarkable increase in the predicted binding energy compared with **DR16** (−55.89 kcal/mol). The binding mode of **DR16.1** inside T3-twiOO is presented in Figure 4B,C. **DR16.1** establishes a hydrogen bond between the NH at the 3-F-phenol of the ligand and the backbone CO of Thr198. A π–π stacking interaction was identified between the benzofuran of the ligand and the residue Phe125 of TASK-3 (Figure 4B,C), an interaction that was previously identified for **DR16** (Figure 3A,B). The 3-F-phenol group of **DR16.1** is oriented towards the fenestration, establishing hydrophobic contacts with the residues at the entrance of this cavity, and the 4-hydroxybenzamide is oriented to the central cavity.

Subsequently, we outsourced the synthesis of compound **DR16.1** to AKos Consulting and Solutions GmbH (http://www.akosgmbh.de/akosgmbh.html). **DR16.1** was obtained as a white solid; m.p: (146.0–148.9) °C; IR-FT (KBr, cm^−1^): 3345, 1648, 1626, 1429; ^1^H NMR (400 MHz, DMSO) δ 10.12 (s, 1H), 9.99 (s, 1H), 9.78 (s, 1H), 9.67 (s, 1H), 7.90 (d, J = 8.7 Hz, 1H), 7.43 (s, J = 6.7 Hz, 1H), 7.41 (d, J = 1.8 Hz, 1H), 7.38 (d, J = 8.2 Hz, 1H), 7.20 (t, J = 7.7 Hz, 1H), 6.87 (d, J = 8.7 Hz, 1H), 6.77 (s, 1H), 6.59 (dd, J = 12.2, 2.5 Hz, 1H), 6.54 (dd, J = 8.6, 2.6 Hz, 1H), 4.24 – 4.02 (m, 1H), 1.53 (d, J = 7.1 Hz, 1H); ^13^C NMR (100 MHz, DMSO) δ 170.36, 165.44, 161.11, 158.56, 157.19, 156.37, 156.26, 154.76, 148.20, 130.33, 129.73, 127.29, 125.24, 123.25, 120.55, 118.06, 117.24, 117.12, 115.42, 111.36, 103.59, 103.29, 103.07, 49.07, 16.51; (ESI, *m*/*z*): calculated for C_24_H_19_FN_2_O_5_^+^ [M]^+^ 434.1278 found [M + Na]^+^ 456.0063. In Appendix A, all spectrum for the characterization of **DR16.1**, as well as the assignment for protons and carbons signals, are shown.

The biological activity of this novel compound against TASK channels was determined (Figure 5), presenting IC_50_ values of 14.17 ± 3.4 and 21.21 ± 5.5 µM against TASK-3 and TASK-1, respectively. **DR16.1** is four-fold more active against TASK-3 than **DR16**, while the activity against TASK-1 is almost the same.

### 2.7. ADME Prediction

Physicochemical descriptors were calculated for **DR16** and **DR16.1**, including molecular weight (MW), total number of hydrogen bond donors (HB-D), total number of hydrogen bond acceptors (HB-A), rotatable bonds, total solvent accessible surface area (SASA), total solvent-accessible volume (MV), and van der Waals surface area of polar nitrogen and oxygen atoms (PSA). In addition, pharmacokinetics properties for both inhibitors, such as logP (octanol/water), logKp for skin permeability, percentage of human oral absorption in the gastrointestinal system, and violations according to Lipinski’s rule of five [36,37], were also determined. The predicted properties are listed in Table 5 and Table 6. Both compounds have an MW < 500 g/mol, which is optimal for a potential drug, and all the calculated physicochemical descriptors and pharmacokinetics properties are in the defined acceptable range to evaluate drug-likeness. The theoretical findings presented here together with the biological activity measurements indicate that our reported compounds have a specific pharmacological activity and have properties that would likely make them orally appropriate for humans.

## 3. Discussion

There has been no information of computational screening targeting the central cavity and/or fenestrations of K_2P_ channels. Recently, Luo et. al. developed a virtual screening but in the extracellular cap of K_2P_ channels to identify inhibitors targeting this site [38]. Nevertheless, due to their pharmacological potential as protein targets in diverse diseases, much evidence has been recently accumulated regarding the molecular characteristics underlying the interactions between different compounds and K_2P_ channels [39].

In the current work, we proposed a protocol that includes pharmacophore-based virtual screening, docking-based high throughput virtual screening, re-docking to refine poses, and binding free energy calculations to find new inhibitors for TASK-3 channels. The success in the finding of hit compounds (DR1 to DR19) indicates that our assumptions in the ‘*e*-Pharmacophore hypothesis in conjunction with the binding in our four TASK-3 models’ (derived by using information of the known blockers) were correct and useful for the identification of novel active compounds (e.g., **DR16**), whose activity can be further optimized (e.g., **DR16.1**).

We also identified the putative residues involved in the binding site of the studied compounds and our results contain some residues of the previously identified A1899 [18,30,35] and PK-THPP (compound 23 in this study) [20,40] binding sites. **DR16** and **DR16.1** share 11 residues in their putative binding site in TASK-3 channels and 63.6% of these residues belong to the binding site of A1899. Within this 63.6%, three residues are also part of the PK-THPP binding site (Appendix A). However, a mutagenesis study is mandatory to confirm the binding site of these new drugs. 

Since it is not possible to know a priori the preferred conformation for including TASK-3 modulators, both closed and opened states of the channel, in the different TASK-3 homology models generated (Table 1, Appendix A), were considered. Analysis of the different fenestration states in T3-trCO (Appendix A) shows how the hydrophobic interactions between Leu239 (TM4 segment) with Leu197 (TM2 segment) and Val115 (inner helix 1), as well as between Ile235 (TM4 segment) with Leu197 and Val115, modulate the fenestration opening–closing mechanism. These interactions are in concordance with the results presented by Brohawn et al. [14], where the residues Leu151, Leu236, Ile279, and Leu283 of TRAAK are implicated in the opening–closing mechanism of the TRAAK fenestration [41] (TASK-3 residues Val115, Leu197, Ile235, and Leu239 are equivalent to TRAAK residues Leu151, Leu236, Ile279, and Leu283, respectively). As observed on previously reported structures of K_2P_ channels, TASK-3 fenestrations are cavities formed by hydrophobic residues [41,42,43], and the hydrophobic regions of the ligands can be included in these cavities. In our report, ligands **DR16** and **DR16.1** establish hydrophobic interactions with Leu239 and Ile235 at the fenestrations. Also, **DR16** interacts hydrophobically with Val115 and through a hydrogen bond with Leu 197 (Figure 3D). With Leu197, **DR16.1** established hydrophobic interactions (Figure 4B). The mutagenesis study previously suggested could reveal whether these new drugs interfere with the opening/closed mechanism at the fenestrations of TASK channels in a similar way as bupivacaine could do it in TASK-1 [19]. 

The *e*-Pharmacophore model derived from the 12 blockers of Table 2 presents two hydrogen bond acceptors (*A1* and *A2*) and one aromatic ring (*R*) (Figure 1). These results are in concordance with those previously reported by our group [35] since the common pharmacophore identified for TASK-1 and Kv1.5 blockers is similar to our model. However, it differs in the position of the aromatic ring (Figure 6A,B). Also, *A1* and *R* groups are contained in the shared seven-point pharmacophore *RRAHRHA* of the 5,6,7,8 tetrahydropyrido[4,3-d]pyrimidine derivatives (Figure 6B,C). Comparing the three pharmacophores, they will differ because they define the interaction patterns of three different groups of bioactive molecules that interact with TASK channels: Those that interact simultaneously with TASK-1 and K_V_1.5 channels (Figure 6A) [34], those that interact with high affinity with TASK-3 channels (Figure 6C) [20,24], and the pharmacophore reported and used here for TASK-3 blockers with different affinities (Figure 6B). This comparison reflects that a pharmacophore is, definitely, a quantitative measure of molecular similarity. However, some features must be shared between these three groups of molecules, for example, the hydrogen bond acceptor groups that can establish interactions with the threonines of the selectivity filter of TASK channels [20,30].

After pharmacophore identification, *e*-Pharmacophore-based virtual screening (*e*-PBVS) was performed to select which compounds of the ZINC database (>22 million) fitted the model requirements; then, a docking-based HTVS following by a re-docking process with a more precise function and binding free energy calculations (MM-GBSA) were done to finally obtain 19 ligands (DR1–DR19 in Table 4 and Appendix A). These hits contain at least two rigid aromatic units connected by amide or ester groups (except DR7), which act as linkers. They also have H-bond acceptor groups and hydrophobic groups that could interact with the hydrophobic residues of the TASK-3 binding site. These common chemical features between the obtained hits and the previously reported blockers allowed us to perform the experimental evaluation of the identified compounds. 

The experimental activity evaluation was performed using patch clamp. We obtained one lead ligand from the 19 tested: **DR16** (IC_50_ = 56.8 ± 3.9 μM); this compound binds TASK-3 with a 1000-fold lower affinity with respect to the most active compounds of the THPP series (compound 23) [20,25], the most active compound reported to date. However, its activity is in the same IC_50_ range with respect to other reported blockers, such as dihyro-β-erythromidine, doxapram, GW2974, L-703,606, loratadine, mevastatin, mibefradil, and octoclothepin (Table 2). Our results are in concordance with those reported by several authors where novel modulators have been identified through virtual screening and/or molecular docking simulations [25,44,45,46,47,48,49,50,51], with a successful result in the prediction of compounds with the same biological activity range of the reported modulators used to construct the pharmacophore model. Using **DR16**, we designed a derivative (**DR16.1**) converging the common pharmacophore identified in TASK-3 blockers (Figure 1) and present in **DR16** (Figure 2) as well. This novel inhibitor has an IC_50_ = 14.17 ± 3.4 and 21.21 ± 5.5 µM against TASK-3 and TASK-1, respectively. Both inhibitors, **DR16** and **DR16.1**, presented ADME/tox (Administration, Distribution, Metabolism, Excretion and Toxicity) properties in accepted ranges for druggability, which suggests that scaffold modifications of these molecules can lead to drug-like compounds. Thus, these two compounds are part of the reduced number of K_2P_ channel modulators reported until today and they are novel scaffolds that could be chemically optimized in the future to get more potent TASK modulators. **DR16.1** activity is four times better than **DR16** activity. The putative binding site of **DR16.1** (Figure 4B) exhibits 11 different residues from the putative binding site of **DR16**. Within them, more than 50% are residues of the binding site of high affinity compounds, A1899 and PK-THPP (Appendix A). Another factor that could increase the activity of **DR16.1** in TASK-3 could be the hydrogen bond established with Thr198, which is not present in **DR16**. Thr198 is at the S4 site of the selectivity filter and we observed recently that the marked difference in the potency of compounds of THPP series (blocking TASK-3 in the nanomolar range), with respect to some less potent compounds of the series (inhibiting TASK-3 channels in the micromolar range), is due to the presence of a hydrogen bond interaction with a threonine of the selectivity filter [20]. Hydrogen bond interactions with threonines of the selectivity filter are also essential for A1899 binding in the TASK-1 channel [18]. 

The experimental activity also reveals that **DR16.1** increased its activity four times in TASK-3 but not in TASK-1 with respect to **DR16**. We hypothesize that this is due to the interactions that appear in **DR16.1** with residues of the binding site of high-affinity TASK-3 compounds, such as PK-THPP, that are not present in **DR16** (i.e., Gln126, Ala237, Appendix A). 

The results presented in this work allow us to conclude that interactions of the identified compounds could be established in the TASK-3 inner cavity and the fenestrations. However, a mutagenesis approach is certainly needed to discern the role of both cavities in the binding sites. **DR16** shows a binding similar to norfluoxetine and bupivacaine, which are located in the fenestrations of TREK-2 [16] and TASK-1 [19], respectively. **DR16.1** orientation is similar to THPP in TASK-3 [20] or A1899 in TASK-1 [18]. 

## 4. Materials and Methods

### 4.1. TASK-3 Modeling

Since the structure of TASK-3 has not been solved, four homology models were developed using the following crystal structures as templates: TRAAK (PDBs:4I9W and 3UM7), TREK-2 (PDB:4BW5), and TWIK-1 (PDB:3UKM). These structures have differences in the fenestration states (they could be open or closed); therefore, different TASK-3 models were used to study the interactions between the lead ligands and TASK-3 with diverse fenestrations characteristics. The steps of the TASK-3 modeling are represented in stage 1 of the general flowchart (Scheme 1). The TASK-3 homology models were built and optimized using ICM software v3.8 [52]. A multiple alignment was used [5] to align the template sequences with the TASK-3 sequence. The four models were named according to the template and the fenestration state (Table 1). In this sense, the model names are: T3-treCC (TASK-3 built from TREK-2 in Close-Close fenestration state), T3-twiOO (TASK-3 built from TWIK-1 in Open-Open fenestration state), T3-trCO (TASK-3 built from TRAAK in Close-Open fenestration state), and T3-trOO (TASK-3 built from TRAAK in Open-Open fenestration state). 

Maestro v.9.2 software (Schrödinger, LLC, New York, NY, USA, 2011) was used to prepare the systems, specifically for the addition of hydrogen atoms and for assignment of the bond order and partial charges to the homology models. Then, they were embedded into a pre-equilibrated phosphatidyl oleoyl phosphatidylcholine (POPC) bilayer in a periodic boundary condition box (15 × 15 × 15 Å^3^) with pre-equilibrated single point charge (SPC) water molecules. Two K^+^ ions were associated to the models at positions S2 and S4, and two water molecules at sites S1 and S3 of the selectivity filter. Finally, the systems were neutralized by adding K^+^ counter ions to balance the net charge of the systems and KCl at a concentration of 0.096 M was added to simulate physiological conditions of the channel. An excluded region for counter ions was set at 5 Å from the selectivity filter of the models. 

The constructed models were subjected to molecular dynamics (MD) simulations to reduce any close contacts resulting from the inclusion of new residues. All the MD calculations were performed using the OPLS-AA force field [53] within the Desmond package v2.0 [54] contained in the Maestro 9.2 suite. The simulation was set in 10 ns and an isothermal-isobaric ensemble, with the temperature (300 K), pressure (1 atm) and number of atoms constant using the Nosé–Hoover method with a relaxation time of 1 ps applying the MTK algorithm. The SHAKE algorithm [55] was employed for every hydrogen atom and the cutoff for van der Waals forces was set at 9 Å and the long-range electrostatic forces were modeled using the particle mesh Ewald method. A restriction was applied on the backbone atoms of the protein and K^+^ ions in the selectivity filter with a spring constant of 0.5 kcal × mol^−1^ × Å^−2^. Data were collected every 2 fs during the MDs. The stability of the models during the MDs was validated by calculating the RMSD, and the quality (after MDs) was validated using PROCHECK [28] and ProSA [29].

### 4.2. e-Pharmacophore Modeling

Twelve diverse TASK-3 modulators (Table 2) were taken for hypothesis generation using an energy-optimized pharmacophore (*e*-Pharmacophore). *e*-Pharmacophore is an approach to generate structure-based pharmacophores [56], which utilizes a scoring function to accurately characterize protein–ligand interactions, resulting in improved database screening. The structures were sketched and processed using LigPrep with the force field OPLS-2005 [57]; possible states of ionization at pH 7.0 ± 2.0 were generated with Epik. 

*e*-Pharmacophore and ligand mapping were generated from TASK-3 modulators selected for the present study (Table 2) with the software Phase [58], using six pharmacophore features: Hydrogen bond acceptor (*A*), hydrogen bond donor (*D*), hydrophobic group (*H*), negatively charged group (*N*), positively charged group (*P*), and aromatic ring (*R*). The most active and least active ligands were defined randomly by Phase to develop the *e*-Pharmacophore model. Stereochemical properties as the isomerism were preserved according with the data reported in the literature (Table 2).

In the scoring hypotheses process, all common pharmacophores were examined. From the data set, we chose as active ligands those that have an IC_50_ value under 70 μM (12f, 17e, 23, A1899, Doxapram, GW2974, L-703,606, Loratadine, and Mibefradil); these ligands were aligned to the hypotheses and Phase calculated the score for the actives. Keeping that in mind, the model was subjected to a scoring and rescoring of the less active ligands (Dihydro-beta-erythrodine, Mevastatin, and Octoclothepin). Then, several generated hypotheses were clustered, with the average linkage method [59] producing clusters when the distance between them was the average distance between all pairs of objects in the two clusters.

Finally, the *e*-Pharmacophore features were correlated with the most active TASK-3 modulator (compound 23, Table 2), and a molecular docking using Glide software [60] was done in each TASK-3 model to localize the conserved *e*-Pharmacophore into the inner cavity and fenestrations, where blockers, such as compounds 23 and A1899 (Table 2), interact with TASK-3 [40] and TASK-1 [18,30] respectively, obtaining four complexes, corresponding to each TASK-3 model with the *e*-Pharmacophore (molecular docking 1 process in Scheme 1). 

### 4.3. Virtual Screening

The hypothesis generated in the *e*-Pharmacophore mapping step, docked into the last frame of each TASK-3 model MDs, was used as a query for screening within the ZINCPharmer pharmacophore software [33]. This software searches a database of conformations calculated from the purchasable compounds of ZINC database [34]. In total, 215,407,196 conformations from 22,723,923 compounds were subjected to an *e*-Pharmacophore-based virtual screening (*e*-PBVS) process using ZINCPharmer (molecular docking 2 process in Scheme 1). Hits were filtered by setting 1 as the maximum limit of hits per conformation, 1 as a maximum hits per molecule, 1 as the number of varied orientations of different conformations returned for each molecule, and the best 5000 hits were considered. The hits were also filtered by setting 1 Å as the maximum RMSD to restrict the hits to those that have the best overall geometric match to the *e*-Pharmacophore hypothesis [61]. The *e*-PBVS was performed for each model (the four models described above), resulting in 5000 hits for each TASK-3 model, with a total of 20,000 hits derived from the *e*-PBVS.

The 5000 hits were re-screened by high-throughput virtual screening (HTVS) (molecular docking 3 process in Scheme 1) using the virtual screening workflow implemented in Maestro version 9.2. All molecules were prepared using LigPrep. The pre-filtering process was done by Lipinski’s rules [37] and the filtering process was done using QikProp [62] as described elsewhere [63,64]. The four constructed TASK-3 models were employed as receptors of the docking steps. The grid box was center in the inner cavity under the selectivity filter; the dimensions in each model were 35 × 35 × 35 Å^3^ to cover the inner cavity and the fenestrations (Appendix A). The residues L122 and L239, identified as part of compound 23’s binding site [40], were included in the grid. The HTVS was done with Glide [60] using the HTVS scoring function to estimate protein–ligand interaction affinities. The results of the HTVS were filtered according to the Glide score; post-docking minimization was done with OPLS-2005 force field to optimize the ligands’ geometries. A total of 2000 hits were obtained with this protocol, 500 hits for each TASK-3 model.

The 500 hits for each TASK-3 model (2000 hits in total) were then subjected to the re-docking process (molecular docking 4 process in Scheme 1) with Glide XP (extra-precision algorithm) [60]. In total, 400 hits were obtained with the re-docking protocol, 100 hits for each TASK-3 model. 

### 4.4. Binding Free Energy Calculations

The computational method of Molecular Mechanics-Generalized Born Surface Area (MM-GBSA), which combines molecular mechanics energy and implicit solvation models [65], was employed using Prime [53] after the re-docking process to rescore and analyze the 100 hits from the ZINC database corresponding to each TASK-3 model. In MM-GBSA, the binding free energies between ligands and receptors (TASK-3 models), used to generate the complexes, were calculated as:(1)$$\Delta G_{bind}=\Delta H-T\Delta S\approx \Delta E_{MM}+\Delta G_{sol}-T\Delta S,$$
(2)$$\Delta E_{MM}=\Delta E_{internal}+\Delta E_{electrostatic}+\Delta E_{vdw};\;\Delta G_{sol}=\Delta G_{PB/GB}+\Delta G_{SA},$$
where $\Delta E_{MM}$, $\Delta G_{sol}$, and −TΔS are the changes in the molecular mechanics energy, solvation-free energy, and conformational entropy upon binding, respectively. $\Delta E_{MM}$ includes $\Delta E_{internal}$ (bond, angle and dihedral energies), electrostatic, and van der Waals energies. $\Delta G_{sol}$ is the sum of the electrostatic solvation energy, $\Delta G_{PB/GB}$ (polar contribution), and non-electrostatic solvation component, $\Delta G_{SA}$(non-polar contribution). The polar contribution was calculated by using the generalized born model, while the non-polar energy was calculated by the solvent accessible surface area (SASA) [66,67].

The VSGB solvation model [68] and OPLS-2005 force field were employed to accomplish the calculations. Residues located at 5 Å from the ligands were included in the flexible region, and all other protein atoms were kept frozen.

### 4.5. ADME Prediction

The absorption, distribution, metabolism, and excretion (ADME) properties of **DR16** and **DR16.1** were predicted by using the program QikProp [62,69]. With this software, some significant physicochemical descriptors and pharmaceutical properties were also predicted. The program was processed with the default parameters, and predicted 44 properties for each lead ligand, such as the number of hydrogen bond (H-bond) donors and acceptors, molecular weight, and calculated LogP (octanol/water), among others. The program also evaluated, with the calculated descriptors, the acceptability of the compounds based on Lipinski’s rule of five, which predicts that poor absorption or permeation is more likely when the compound has more than 5 H-bond donors, more than 10 H-bond acceptors, a molecular weight greater than 500 g/mol, and a calculated LogP greater than 5 [36].

### 4.6. Heterologous TASK-3 Expression and Electrophysiological Screening (Patch-Clamp)

HEK-293 cells were maintained in DMEM-F12 media supplemented with 10% FBS and 1% penicillin/streptomycin. Transient transfections (1 µg plasmid) were done with a DNA ratio of 3:1 (plasmid encoding TASK-3 channel (GenBank accession NP_057685.1): Plasmid encoding for green fluorescent protein as marker) using Xfect^TM^ polymer (Takara Bio Company, Kanagawa, Japan). Whole cell recordings were performed at room temperature 24 h post-transfection using a PC-501A patch clamp amplifier (Warner Instruments, LLC, Hamden, CT, USA) and borosilicate pipettes as described elsewhere [70,71]. Recording pipettes were filled with an intracellular solution contained (in mM): 145 KCl, 5 EGTA, 2 MgCl_2_, 10 HEPES, adjusted to pH 7.4 with KOH. Recording solution containing (in mM): 135 NaCl, 5 KCl, 1 MgCl_2_, 1 CaCl_2_, 10 HEPES, 10 Sucrose, adjusted to pH 7.4 with NaOH. Ion channel currents were measured with a voltage protocol (400-ms steps from –100 mV to 100 mV with an increment of 10 mV and a holding potential of –80 mV). The TASK-3 blockade was analyzed at +80 mV test pulse. 

The hit compounds identified from the ZINC database and selected for the experimental activity evaluations were acquired from the following suppliers: AKos Consulting & Solutions Deutschland GmbH (Steinen, Germany), Ambinter c/o Greenpharma (Orléans, France), EnamineStore Ltd. (Kyiv, Ukraine), and Vitas-M Limited (Hong Kong, China). All the compounds were prepared by directly dissolving them in external bath solution (recording solution) to obtain the desired final concentrations.

### 4.7. Chemical Characterization of New TASK-3 Blockers

The new molecules designed in this work were characterized by spectral data (IR, ^1^H, and ^13^C NMR mono and two-dimensional, MS). ^1^H and ^13^C NMR spectra (400 MHz for ^1^H and 100 MHz for ^13^C) were recorded on an AM-400 spectrometer (Bruker, Rheinstetten, Germany), using DMSO-d_6_ as solvents. The chemical shift (δ) of solvent is reported at 2.5 for ^1^H-NMR and 40.09 for ^13^C-NMR, δ and *J* values are reported in ppm and Hz, respectively. The signals are represented as follow: Singlet (s), doublet (d), triplet (t), and multiplet (m). IR spectra (KBr pellets, 500–4000 cm^−1^) were obtained from a NEXUS 670 FT-IR spectrometer (Thermo Nicolet, Madison, WI, USA). High-resolution mass spectra (HRMS-ESI) were obtained from a Thermo Fisher Scientific Exactive Plus mass spectrometer. The analysis was performed at a heater temperature of 50 °C, sheath gas flow of 5, sweep gas flow rate of 0, and spray voltage of 3.0 kV in positive mode. The accurate mass measurements were performed at a resolution of 140,000. Melting points (uncorrected) were determined on an Electrothermal IA9100 melting point apparatus (Stone, Staffs, UK).

## 5. Conclusions

The successful identification of new TASK-3 modulators through pharmacophore modeling, virtual screening protocol, and rational design confirms the usefulness of the identified pharmacophore hypothesis for the design of TASK-3 blockers. We found a simple pattern, which was combined with models derived from X-ray crystallographic structures. It is known in the literature that homology models are imprecise by definition; however, we found useful information from them because four variants of models were considered. These variants considered relevant different conformations, which were described well for K_2P_ channels, which influence the size and electrostatic properties of the studied biding sites. We believe that this consideration was essential for the success of our protocol. 

An additional point is that our report proposes the binding mode of the studied compounds at the fenestrations (**DR16**) and the inner cavity and the fenestration (**DR16.1**) of the channel. These are relevant insights into the binding mode of new TASK-3 modulators that must be checked by site-directed mutagenesis. Although the identified ligand **DR16** and designed compound **DR16.1** showed moderate potencies, the conserved pharmacophore and novel chemical characteristics of this chemical class make them good candidates for future development into highly potent TASK-3 modulators through medicinal chemistry optimization.

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
