# Peer review of "Discovery of Novel TASK-3 Channel Blockers Using a Pharmacophore-Based Virtual Screening"

_ijms, 2019, doi:10.3390/ijms20164014_

Round 1

Reviewer 1 Report

The paper presents a comprehensive molecular modeling and chemical study to the search for new TASK-3 inhibitors. Various molecular modeling approaches were used; however, the compounds found in the study have low activity (IC50 higher than 1000 nM). Morevoer, retrospetive study for homology models evaluation needs to be performed to have awareness of their potential in distinguishment between active and inactive compounds

Author Response

We thank your comment. The compounds found in our study were the result of a pharmacophore-based virtual screening, which is a method used for finding hit compounds, not lead compounds (about hit and lead compounds see reference: Nat Rev Drug Discov 2003, 2, 369–378). A hit compound is a molecule that shows the desired type of activity in a screening assay (not necessarily a very potent activity), and in our case, we identified TASK-3 channel blockers (hits). It is important to develop ‘pharmacologically relevant screening experiments’ for hit discovery (our compounds) and hit compounds can be used in subsequent hit-to-lead optimization process. Lead compounds (more potent compounds derived from the identified hits) could be generated by transforming hits in a chemical lab (medicinal chemistry and structure-activity relationships). Therefore, identification of hit compounds can be done by using virtual screening (or high-throughput screening) and development of lead compounds is commonly done in a medicinal chemistry lab.

After checking these definitions, Reviewer 1 should consider that:

1- We found hit compounds which is the first step for finding more potent compounds. This endeavor is not trivial by considering that we explored a cavity of a potassium channel, which is more difficult than a search in an enzyme cavity.

2- The success in the finding of hit compounds indicates that our modeling protocol is a ‘pharmacologically relevant screening experiment’. At the same time, it indicates that other researchers could reproduce our protocol to find novel TASK-3 blockers. Reviewer should consider that we found one active compound from an initial dataset containing millions of compounds.

3- The success in the finding of hit compounds indicates that our assumptions in the ‘e-Pharmacophore hypothesis in conjunction with the binding in our four TASK-3 models’ (derived by using information of the known blockers) were correct and useful for the identification of novel active compounds.

The potential of our models (homology models + pharmacophore) to find novel active compounds (hit compounds) was demonstrated in our report. We should have been very lucky to have found a lead compound with a pharmacophore-based virtual screening.

Despite these considerations, in the revised version (line 311 – 315) we included a sentence indicating the value of our approach.

Reviewer 2 Report

The manuscript entitled “Discovery of novel TASK-3 channel blockers using a pharmacophore-based virtual screening”, describes a molecular modelling study, what includes: pharmacophore modelling, molecular docking studies and free energy calculations. With these tools, the authors aim to address search and design new blockers of TASK-3 channels.

The subject of this research is very interesting, although I suggest a minor improvement, especially in results and discussion description, as well as a more detailed and structured writing.

The manuscript is not properly organized or structured. For example, the scheme 1, named on line 100 (page 3), is shown on page 14, or DR16 and DR16.1 ligands, are shown on page 7 and 9, respectively. From my point of view, this would be solved by starting the manuscript with materials and methods descriptions.

The goal is to find new heterocyclic derivatives with potential utility in these potassium channels. However, the proposal could be improved if it is not limited to such structures, since, for example, compound A1899 is a potent and highly selective blocker of the K (2P) channels and without heterocycles.

The pharmacophore modelling is approached using 12 ligands compiled from the literature on this membrane proteins (K2P9.1 channels). The broad ranges of activity (IC50: 0.035-159 mM) are surprising, as well as the reported activity of the ligand GW2974 (IC50: 50.1 ± 41.8, reference 32), or even that some ligands are designated as inactive ligands (line 446, pag. 15). In this sense, the design should be improved, taking into account also that pharmacophoric models constructed from so few ligands are not maintained.

The molecular docking studies are developed from homology models which are built and optimized using ICM software. The ICM version used should be referred. In these models, protein templates with low sequence identity values are used. It is known that in order to ensure good homology models, they must be constructed from protein templates with a sequence identity of 30% or more. However, sequence identities used here are lower than 30%. This fact must be justified.

After the virtual screening, 19 ligands are proposed as possible hits of TASK-3 channels. It can be deduced that only one of them (DR16,) shows moderate activity? This fact could call into question the validity of virtual screening procedure described, since the success rate would be 5%.

A study of the ADME properties of the lead ligands is included at the end of the manuscript. The moderate activity values of these compounds do not justify include druglikeness studies, which are more oriented to compounds which might show therapeutic potential.

As a last comment, for the design of the DR16.1 ligand, the need to use all the computational tools described in this research could be doubtful.

Author Response

We thank your valuable comments. Below you will find our answers

1. The manuscript entitled “Discovery of novel TASK-3 channel blockers using a pharmacophore-based virtual screening”, describes a molecular modelling study, what includes: pharmacophore modelling, molecular docking studies and free energy calculations. With these tools, the authors aim to address search and design new blockers of TASK-3 channels.

The subject of this research is very interesting, although I suggest a minor improvement, especially in results and discussion description, as well as a more detailed and structured writing.

The manuscript is not properly organized or structured. For example, the scheme 1, named on line 100 (page 3), is shown on page 14, or DR16 and DR16.1 ligands, are shown on page 7 and 9, respectively. From my point of view, this would be solved by starting the manuscript with materials and methods descriptions.

R/ We thank your comments and suggestions. We moved the ‘Scheme 1’ to page 3 and changed the final lines of the introduction section in order to mention DR16 and DR16.1 in the result section (page 7 and 8). We would also like to mention that it is not possible to start the manuscript with the material and methods sections because the IJMS guidelines require the results been presented first and then the materials and methods.

2. The goal is to find new heterocyclic derivatives with potential utility in these potassium channels. However, the proposal could be improved if it is not limited to such structures, since, for example, compound A1899 is a potent and highly selective blocker of the K (2P) channels and without heterocycles.

R/ We thank your comments and suggestions. We made a typo in the Introduction section. Now lines 87 reads as follows: “Therefore, it is necessary the development of simple theoretical/experimental methodologies to find new compounds with different chemical nature with potential usefulness as TASK-3 modulators.

3. The pharmacophore modelling is approached using 12 ligands compiled from the literature on this membrane proteins (K2P9.1 channels). The broad ranges of activity (IC50: 0.035-159 mM) are surprising, as well as the reported activity of the ligand GW2974 (IC50: 50.1 ± 41.8, reference 32), or even that some ligands are designated as inactive ligands (line 446, pag. 15). In this sense, the design should be improved, taking into account also that pharmacophoric models constructed from so few ligands are not maintained.

R/ The most important characteristics for a virtual screening are the ‘diversity’ of the compounds used for constructing the model and the fact that these compounds must be ‘active’ (at nM or microM level) against the selected target. Compounds GW2974, dihydro-Beta-erythroidine, octoclothepin and mevastatin (with the lowest activity in the dataset, 159 microM) are among the 12 ligands used for the virtual screening because they provide diversity to our search. These compounds have ‘low activity’, although, they are all TASK-3 modulators because they share the pharmacophore identified in our work.

We are aware that the term ‘inactive’ is confusing in our protocol; therefore, in the revised version (section 4.2) we changed this term by ‘less active’.

4. The molecular docking studies are developed from homology models which are built and optimized using ICM software. The ICM version used should be referred. In these models, protein templates with low sequence identity values are used. It is known that in order to ensure good homology models, they must be constructed from protein templates with a sequence identity of 30% or more. However, sequence identities used here are lower than 30%. This fact must be justified.

R/ Thanks for your comment. The ICM version used is now referred (line 409).

Regarding the low sequence identity, we are aware that our sequence identity is lower that 30% when the entire channel sequences are compared (TASK-3 aminoacidic sequence shares 27.2% identity with TWIK-1, 23.7% with TRAAK, and 26.2% with TREK-2). However, if we eliminate the C- and N-terminal region of the query (TASK-3) and the templates and compare the transmembrane sections (considering that C- and N-terminal region are not part of the described TASK-3 active site à REF 6, 18-20, 30) and perform BLAST analysis between TASK-3 and TWIK-1, for example, we get a 33% sequence identity. Therefore, the models are in an acceptable range of sequence identity to be used as templates to build comparative models of membrane proteins (REF 29: Forrest, L. R.; Tang, C. L.; Honig, B. On the accuracy of homology modeling and sequence alignment methods applied to membrane proteins. Biophys. J. 2006, 91, 508–517) In addition, it is quite common to use templates under 30% sequence identity to build comparative models of ion channels due the low availability of ion channels crystal structures (about ion channels homology models see references: i) Anishkin, A., et al. (2010). Symmetryrestrained molecular dynamics simulations improve homology models of potassium channels. Proteins: Structure, Function, and Bioinformatics, 78(4), 932-949., ii) Giorgetti, A., & Carloni, P. (2003). Molecular modeling of ion channels: structural predictions. Current opinion in chemical biology, 7(1), 150-156.; iii) Ravna, A. W., & Sylte, I. (2011). Homology modeling of transporter proteins (carriers and ion channels). In Homology Modeling (pp. 281-299). Humana Press.)

5.  After the virtual screening, 19 ligands are proposed as possible hits of TASK-3 channels. It can be deduced that only one of them (DR16,) shows moderate activity? This fact could call into question the validity of virtual screening procedure described, since the success rate would be 5%.

R/ This result is outstanding in virtual screening. It is more common to find maybe one compound from hundreds of proposed compounds, or it is even possible not to find any active compound in experiments. A virtual screening is successful when at least one hit compound is discovered. Thus, it should be considered that virtual screening is an in silico method to guide experiments. In pharmaceutical industry, virtual screening could be successful after testing millions of compounds when a few hit compounds are identified.

It is common to have a success rate below 5%. Only one compound demonstrates that our pharmacophore model is a useful filter since it identifies one TASK-3 modulator from an initial dataset containing millions of compounds. This single compound was identified with the help of a theoretical model, without testing millions of compounds in experiments.

6.  A study of the ADME properties of the lead ligands is included at the end of the manuscript. The moderate activity values of these compounds do not justify include druglikeness studies, which are more oriented to compounds which might show therapeutic potential.

R/ It is pertinent to have a report of predicted ADME properties for future development of compounds derived from DR16 and DR16.1. These values demonstrate that DR16 and DR16.1 scaffold modifications can lead to drug-like compounds. In the revised version (line 378-379) a sentence indicating the relevance of ADME results for future structure optimization is included.

7.  As a last comment, for the design of the DR16.1 ligand, the need to use all the computational tools described in this research could be doubtful.

R/ DR16.1 is a derivative of DR16 and DR16 was found by using our virtual screening protocol; therefore, the need to use all the computational tools described in this research for the design of DR16.1 are well justified.